# Robot-Gated Interactive Imitation Learning with Adaptive Intervention Mechanism

**Haoyuan Cai** [1]  **Zhenghao Peng** [1]  **Bolei Zhou** [1]

## Abstract

Interactive Imitation Learning (IIL) allows agents to acquire desired behaviors through human interventions, but current methods impose high cognitive demands on human supervisors. We propose the *Adaptive Intervention Mechanism* (AIM), a novel robot-gated IIL algorithm that learns an adaptive criterion for requesting human demonstrations. AIM utilizes a proxy Q-function to mimic the human intervention rule and adjusts intervention requests based on the alignment between agent and human actions. By assigning high Q-values when the agent deviates from the expert and decreasing these values as the agent becomes proficient, the proxy Q-function enables the agent to assess the real-time alignment with the expert and request assistance when needed. Our expert-in-the-loop experiments reveal that AIM significantly reduces expert monitoring efforts in both continuous and discrete control tasks. Compared to the uncertainty-based baseline Thrifty-DAgger, our method achieves a 40% improvement in terms of human take-over cost and learning efficiency. Furthermore, AIM effectively identifies safety-critical states for expert assistance, thereby collecting higher-quality expert demonstrations and reducing overall expert data and environment interactions needed. Code and demo video are available at https://github.com/metadriverse/AIM.

## 1. Introduction

Reinforcement learning (RL) has been successful in a wide variety of tasks, from playing Atari games (Mnih et al., 2013) to autonomous driving (Li et al., 2022a) and robotic

---
[1]Computer Science Department, University of California, Los Angeles (UCLA). Correspondence to: Bolei Zhou <bolei@cs.ucla.edu>.

*Proceedings of the 42nd International Conference on Machine Learning*, Vancouver, Canada. PMLR 267, 2025. Copyright 2025 by the author(s).

### Robot-Gated Interactive Imitation Learning

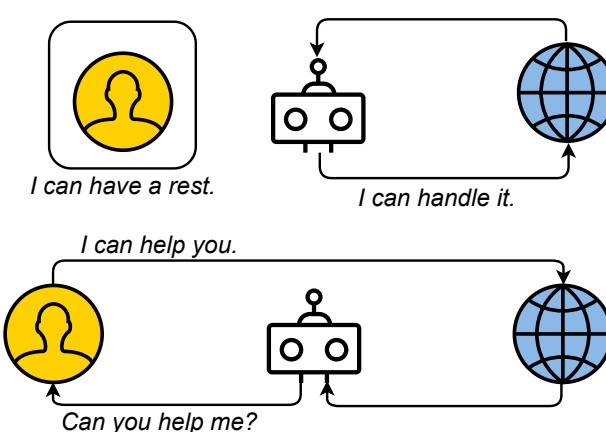

*Figure 1.* In robot-gated IIL, the agent explores the environment and requests human assistance based on an intervention criterion. The expert is not involved in learning until the agent requests help.

manipulation tasks (Finn et al., 2017; Fang et al., 2019; Ganapathi et al., 2021). However, RL often suffers from poor sample efficiency during policy learning, and the reward function struggles to perfectly capture human preferences, causing learned policies to produce unintended behaviors. To address these limitations, researchers have turned to Imitation Learning (IL), which leverages offline expert demonstrations and aims to match human behavior on this offline dataset (Osa et al., 2018). Nevertheless, IL is susceptible to distribution shifts and out-of-distribution (OOD) states, where the agent encounters scenarios not covered by the training data (Ross et al., 2011; Ravichandar et al., 2020; Chernova & Thomaz, 2022; Zare et al., 2024).

Interactive Imitation Learning (IIL) incorporates human participants to intervene in the training process and provide online demonstrations, which helps improve alignment and learning efficiency (Reddy et al., 2018; Kelly et al., 2019; Spencer et al., 2020; Peng et al., 2024; Seraj et al., 2024). There are two different categories of human involvement: human-gated intervention (Kelly et al., 2019; Li et al., 2022b; Peng et al., 2024) and robot-gated involvement (Menda et al., 2019; Hoque et al., 2021a). Human-gated IIL leverages human abilities to identify the agent's

mistakes, predict future trajectories, and adjust the intervention based on the agent's performance. However, this requires humans to continuously monitor the whole training process and immediately intervene in safety-critical states (Peng et al., 2024; 2025). Monitoring the learning process imposes a significant burden on the human supervisor. In contrast, robot-gated IIL allows the agent to autonomously request assistance based on an intervention criterion, reducing the cognitive load on human supervisors. Prior works on robot-gated IIL use uncertainty-based or preference-based intervention criteria, which may fail to align with human intents in deciding whether to intervene (Hoque et al., 2021a; Menda et al., 2019; Hejna et al., 2023). Common uncertainty-based methods (Menda et al., 2019; Kelly et al., 2019) request human help when the uncertainty estimate is larger than a fixed threshold. Without adjusting the threshold adaptively, the agent keeps requesting expert help even when it has successfully imitated the human expert. Furthermore, these methods require tuning multiple hyperparameters for each task and training an ensemble of policy networks to compute action variance, which harms the learning efficiency (Hoque et al., 2021a).

To solve these issues, we propose the *Adaptive Intervention Mechanism* (AIM), a novel IIL algorithm that learns a robot-gated intervention criterion resembling human-gated intervention mechanisms. First, AIM recovers the underlying intervention mechanism that aligns with human intentions, ensuring that the interventions are critical and effective. By training a proxy Q-function that approximates the human's decisions in intervention, AIM proactively requests human assistance when the agent's policy deviates from the expert. Second, as the agent's policy converges toward the expert's, the proxy Q-values decrease, which automatically reduces the intervention rate over time. In this way, AIM adaptively shrinks its requests for help as proficiency improves, without any hand-tuned schedule (see Fig. 8). Compared with robot-gated IIL baselines, AIM effectively seeks human guidance in safety-critical states. Third, AIM eliminates the need for computationally intensive uncertainty quantification by utilizing a proxy value function loss, enhancing training efficiency and scalability. We evaluate our method AIM in MetaDrive (Li et al., 2022a) and MiniGrid (Chevalier-Boisvert et al., 2018), showing that AIM achieves a higher learning efficiency compared to various baselines in both continuous and discrete action spaces.

We summarize our contributions as follows:

1. We propose the *Adaptive Intervention Mechanism* (AIM), an Interactive Imitation Learning algorithm that proactively requests expert intervention during environment interaction. AIM learns a proxy Q-function to adaptively capture states where the agent's actions diverge from the expert's. AIM leverages the real-time

action difference from the expert to automatically reduce the expert intervention rate as the agent's policy converges toward expert behavior.

2. We evaluate our algorithm in MetaDrive and Mini-Grid environments, demonstrating that AIM requires fewer expert demonstrations and monitoring efforts to achieve near-optimal policies.

3. Experiments show that AIM requires fewer environment interactions and expert data to handle the safety-critical states. The expert demonstrations requested by AIM contain corrective actions in safety-critical states so that they can assist a novice agent in imitating the expert's policy.

## 2. Related Work

**Human-Gated Interactive Imitation Learning** Human-gated IIL algorithms rely on human experts to proactively be involved in the training loop and provide corrective actions at dangerous or repetitive states. Human-Gated DAgger (HG-DAgger) (Kelly et al., 2019) and Intervention Weighted Regression (IWR) (Mandlekar et al., 2020) perform imitation learning on human intervention data. These methods do not leverage data collected by agents or limit the human intervention frequency, harming the sample efficiency. EGPO (Peng et al., 2021) and PVP (Peng et al., 2024; 2025) design proxy cost or value functions to suppress the frequency of human involvement. A recent line of human-gated IIL methods, including CEILING (Celemin & Ruiz-del Solar, 2019) and RLIF (Luo et al., 2023) requires humans to manually assign different weights or preference signals in addition to corrective actions. Still, they require humans to monitor the screen during the entire training process and stay aware of the agent's potential mistakes. Although these human evaluative feedbacks reduce the demonstrations needed, providing these signals can be time-consuming and thus incur more human involvement.

**Robot-Gated Interactive Imitation Learning** Robot-gated IIL methods rely on an intervention mechanism, and the key is to lift the burdens on human experts while obtaining sufficient information for the training of the expert policy. Existing robot-gated methods include Ensemble-DAgger (Menda et al., 2019) and Thrifty-DAgger (Hoque et al., 2021a), which utilize the variance of actions as the uncertainty estimates and assume that the expert should intervene in the states with high uncertainty. However, such heuristic criteria may not align with human intervention strategies, and the uncertainty estimates of dangerous states may fluctuate during training. In addition, prior robot-gated algorithms require hyper-parameter tuning based on the tasks (Hoque et al., 2021b; Biré et al., 2024) or manually specify a desired intervention rate (Zhang & Cho, 2016;

Hoque et al., 2021a), which further aggravates the expert's burden and fails to adjust the intervention criterion as the agent becomes proficient. Confidence-based intervention employs task-specific and hard-coded confidence models to estimate the familiarity of current states, such as the nearest neighbor distance to the states in the training data (Chernova & Veloso, 2009; Saeidi et al., 2018). While intuitive, these confidence models struggle in high-dimensional observation spaces and lack generalization across different tasks. In contrast, AIM trains a proxy Q-function that directly captures human intervention decisions and reduces the intervention frequency as the agent becomes proficient, eliminating the need to manually set hyperparameters. By relying on actual intervention data rather than heuristic confidence measures, AIM also generalizes to high-dimensional observations without requiring task-specific engineering.

## 3. Preliminaries

In this section, we introduce our settings of interactive imitation learning environments. The robot environment is modeled by a Markov Decision Process (MDP) $M := \langle \mathcal{S}, \mathcal{A}, \mathcal{P}, r, \gamma, d_0 \rangle$ with a state space $\mathcal{S}$, an action space $\mathcal{A}$, a state transition function $\mathcal{P} : \mathcal{S} \times \mathcal{A} \to \mathcal{S}$, a reward function $r$, a discount factor $\gamma$, and an initial state distribution $d_0$. For any agent policy $\pi_r(a \mid s) : \mathcal{S} \times \mathcal{A} \to [0, 1]$, the expected cumulative return is $J(\pi_r) = \mathbb{E}_{\tau \sim P_{\pi_r}} [\sum_{t=0}^{\infty} \gamma^t r(s_t, a_t)]$, where $P_{\pi_r}$ is the distribution of trajectories $\tau = (s_0, a_0, s_1, a_1, \ldots)$ induced by $\pi_r, d_0$ and $\mathcal{P}$. The reinforcement learning objective is to learn an agent policy $\pi_r(a \mid s)$ which maximizes $J(\pi_r)$. When $\pi_r$ is deterministic, we denote the function $\mu_r(s)$ that outputs a robot action $a_r$ at state $s$. In this paper, we consider the reward-free setting in which the agent has no access to the task reward function $r(s, a)$.

In imitation learning, the robot learns from human behaviors by imitating the human policy $\pi_h(a \mid s)$. Traditional imitation learning algorithms learn from human expert trajectories $\tau_h \sim P_{\pi_h}$, which may lead to poor performance due to out-of-distribution states (Ross et al., 2011).

Human-gated interactive imitation learning incorporates an interaction mechanism in which the expert applies an intervention criterion $I^{\exp}(s, a_r, a_h) : \mathcal{S} \times \mathcal{A} \times \mathcal{A} \to \{0, 1\}$ to decide whether to take control when the agent outputs action $a_r$ at state $s$. The behavior policy that generates action in the training process follows

$$\pi_b(a|s) = (1 - I^{\exp}(s, a_r, a_h))\delta(a - a_r) \\ + I^{\exp}(s, a_r, a_h)\pi_h(a|s). \tag{1}$$

Here, the $\delta$ function in Eq. 1 denotes the Dirac delta distribution, representing a deterministic action $a_r$. In Eq. 1, $I^{\exp}(s, a_r, a_h)$ works as a human-gated intervention criterion.

In robot-gated IIL, the agent has access to the switch-to-human function $I^r(s, a_r)$ when the agent interacts with the environment itself without access to $a_h$. Once the human expert intervenes at state $s$, the agent uses another continue-with-human function $I^h(s, a_r, a_h)$ to decide whether to stop requesting human help. The first switch-to-human function $I^r(s, a_r) : \mathcal{S} \times \mathcal{A} \to \{0, 1\}$ decides whether to ask the human expert for help when the agent takes action $a_r$ at state $s$. The agent uses the second continue-with-human function $I^h(s, a_r, a_h) : \mathcal{S} \times \mathcal{A} \times \mathcal{A} \to \{0, 1\}$ to decide whether to continue requesting expert help. When $I^h(s, a_r, a_h) = 0$, the agent will inform the expert to stop providing demonstrations and will not request human assistance until $I^r(s', \mu_r(s')) = 1$ again at some future state $s'$.

## 4. Method

In this section, we first provide a motivating example in Sec. 4.1 to show the drawbacks of previous uncertainty-based robot-gated Interactive Imitation Learning (IIL) methods. Then, we introduce our approach of learning an adaptive intervention mechanism in Sec. 4.2 that emulates how humans teach an evolving agent policy. We provide our algorithm pipeline in Sec. 4.3.

### 4.1. Motivating Example

Human-gated IIL algorithms like HG-DAgger (Kelly et al., 2019) and PVP (Peng et al., 2024) require humans to monitor the entire training loop and correct the agent's actions at safety-critical states or when the agent encounters difficulties discovering the optimal strategy, as is shown in Alg. 1. To alleviate the burden on human supervisors by eliminating the need for continuous monitoring, we aim to design a robot-gated criterion $I^r(s, a_r)$ to proactively request human assistance in a manner that imitates humans' intervention mechanism $I^{\exp}(s, a_r, a_h)$ in Alg. 1. Prior works of robot-gated IIL, including Ensemble-DAgger (Menda et al., 2019) and Thrifty-DAgger (Hoque et al., 2021a), assume that the agent makes frequent mistakes at novel states and is more proficient at handling frequently encountered states. Therefore, these methods employ heuristic uncertainty estimates to identify novel states and trigger human intervention if the uncertainty of current states exceeds a pre-defined threshold.

In Fig. 2, we illustrate the drawbacks of uncertainty-based methods by showing the mismatch between the uncertainty estimates and the human-gated intervention mechanism. We use the MetaDrive (Li et al., 2022a) and illustrate the cases when the human expert intervenes from two trajectories. The $y$ axis of the uncertainty estimate is $\text{Var}(a_r) - \varepsilon$, where $\text{Var}(a_r)$ is the variance of agent actions and $\varepsilon$ is the switch-to-human threshold in Ensemble-DAgger (Menda et al., 2019), i.e., human demonstration is requested when

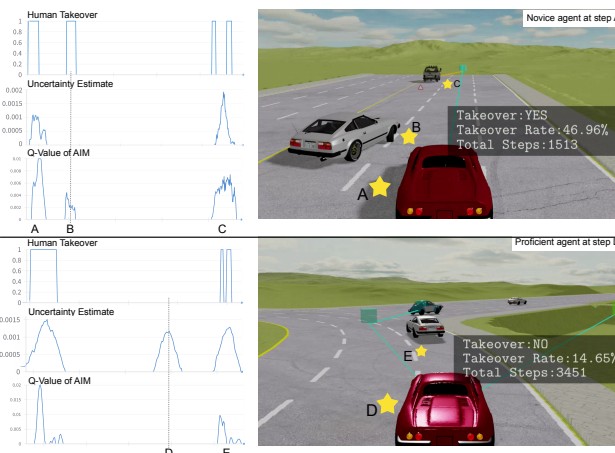

*Figure 2.* Comparison between the uncertainty estimate and our Adaptive Intervention Mechanism (AIM) in interactive imitation learning under MetaDrive safety environments. We plot the human intervention behavior, the corresponding uncertainty estimate, and AIM's Q-value when supervising a novice agent (upper) and a proficient agent (lower). At step B, the front car makes a sudden right turn, and the human believes an intervention is needed. At step D, the expert believes the agent is proficient enough to navigate this curve and avoid collision. This highlights the limitations of fixed uncertainty thresholds and the need for an adaptive intervention mechanism adjusting to the agent's improving performance.

$\text{Var}(a_r) > \varepsilon$. Similarly, we plot $Q_\theta^I(s, a_r) - \beta$ in Fig. 2 to visualize when the AIM agent requests human help, where $Q_\theta^I(s, a_r)$ is AIM's proxy $Q$ function defined in Sec. 4.2 and $\beta$ is a threshold defined in Eq. 6. The curves are smoothed using a 20-step running average.

In steps B and D, the uncertainty estimates fail to align with human intervention decisions: the safety violation at step B is not predicted by the uncertainty estimate, while at step D the uncertainty estimate requests human assistance even when the agent behaves correctly. The uncertainty estimates with a pre-defined threshold limit robot-gated IIL's adaptability to the evolving agent policy. This fact highlights the necessity of adaptive behavior in the design of robot-gated intervention mechanisms. Fig. 8 in the Appendix further contrasts AIM with the uncertainty-based baseline across different training stages. AIM quickly and precisely identifies states where the agent is already competent, while the uncertainty-based method keeps querying the expert at the same locations, wasting intervention effort.

Another problem in robot-gated IIL is how to stop human intervention. Previous works utilize the action difference $\|a_r - a_h\|^2$ as the criterion to determine intervention-stopping function $f(a_r, a_h) = \mathbb{I}[\|a_r - a_h\|^2 > \epsilon]$, where $\epsilon$ is a pre-defined parameter (Menda et al., 2019; Hoque et al., 2021a;b). In addition, some works show that if a hypothetical human expert intervenes and leaves intervention

following the same continue-with-human function

**Algorithm 1** Human-Gated Interactive Imitation Learning

**for** trajectory $i = 1, 2, \ldots$ **do**
  **for** timestep $t = 1, 2, \ldots$ **do**
    Agent samples action $a_r \sim \pi_r(s_t)$.
    Human observes $(s_t, a_r)$ and decides whether to issue an intervention $I^{\exp}(s_t, a_r, a_h)$.
    **if** $I^{\exp}(s_t, a_r, a_h) = 1$ **then**
      Human samples $a_h \sim \pi_h(s_t)$.
      Observe $s_{t+1} \sim \mathcal{P}(\cdot \mid s_t, a_h)$.
      Add $(s_t, a_h, s_{t+1})$ to the human buffer $\mathcal{B}_h$.
    **else**
      Agent takes control.
      Observe $s_{t+1} \sim \mathcal{P}(\cdot \mid s_t, a_r)$.
      Add $(s_t, a_r, s_{t+1})$ to the novice buffer $\mathcal{B}_r$.
    **end if**
    Train $\pi_r$ on $\mathcal{B}_h$ via imitation learning.
  **end for**
**end for**
**Output:** $\pi_r$.

$$f(a_r, a_h) = \mathbb{I}[\|a_r - a_h\|^2 > \epsilon], \qquad (2)$$

then the agent can recover the expert policy with a performance guarantee when $\epsilon$ is small enough (Peng et al., 2021; 2024). Following these works, we set $I^h(s, a_r, a_h) = f(a_r, a_h)$, i.e., the agent stops requesting human for help if $\|a_r - a_h\|^2 \leq \epsilon$.

## 4.2. Training the Adaptive Intervention Mechanism

We propose the *Adaptive Intervention Mechanism* algorithm (AIM), which can recover the humans' underlying intervention criterion and automatically adjust the intervention rate by capturing the evolving nature of the agent policy.

The key idea of AIM is to use a proxy Q-function $Q_\theta^I(s, a_r)$ to approximate the human-gated intervention mechanism: the higher $Q_\theta^I(s, a_r)$ is, the more likely human would intervene. Given a set of human demonstrations $\mathcal{B}_h = \{(s, a_h)\}$ and the current agent policy $\pi_r$, we label the Q value of $Q_\theta^I(s, a_h)$ as $-1$ for any $(s, a_h)$ sampled from $\mathcal{B}_h$, i.e., when the agent takes exactly the action $a_h$ at state $s$, the human participant is unlikely to intervene as the agent aligns with the human policy. In addition, we sample $a_r$ from the current agent policy $\pi_r$ and label the Q value of $Q_\theta^I(s, a_r)$ as $+1$ if $\|a_r - a_h\|^2 > \epsilon$. The expert will likely provide corrective demonstrations if the agent behaves differently, which conforms to Eq. 2 in Sec. 4.1. This learning objective is achieved by fitting the proxy Q-function $Q_\theta^I$ with the following AIM loss:

$$J^{\text{AIM}}(\theta) = \mathop{\mathbb{E}}_{(s,a_h)\sim\mathcal{B}_h}\left[\left|Q_\theta^I(s,a_h)+1\right|^2\right]$$
$$+\mathop{\mathbb{E}}_{(s,a_h)\sim\mathcal{B}_h,a_r\sim\pi_r(s)}\left[f(a_r,a_h)\cdot\left|Q_\theta^I(s,a_r)-1\right|^2\right]. \tag{3}$$

The proposed AIM loss function offers significant advantages over the uncertainty-based intervention criterion in Ensemble-DAgger (Menda et al., 2019) and Thrifty-DAgger (Hoque et al., 2021a) by ensuring adaptability to the evolving agent policy. Uncertainty estimates based on the variance of actions make the agent reluctant to seek humans for help at frequently encountered states, even when the agent behaves differently with humans. In contrast, our proxy Q-function $Q_\theta^I(s,a_r)$ remains responsive to the changing agent policy. Specifically, the Q-value of $Q_\theta^I(s,a_r)$ approaches +1 continuously as long as $a_r$ deviates from $a_h$ at any state $s$ in $\mathcal{B}_h$. This design motivates the agent to request human demonstrations if it keeps making mistakes, mitigating the optimality gaps.

Furthermore, the intervention rate of AIM also adapts to the agent's performance. As $\pi_r$ becomes increasingly aligned with the human policy $\pi_h$, in Eq. 3, a growing number of states $s$ in $\mathcal{B}_h$ will satisfy $f(a_r,a_h)=0$. This leads the average Q-value $Q_\theta^I(s,a_r)$ to decrease towards $-1$, shrinking the intervention rate smoothly, making the agent less likely to request expert help in states that it has already mastered. The gradually decreasing intervention rate aligns with the human intervention mechanism in Fig. 2, where the expert intervenes less frequently as the agent demonstrates proficiency.

In addition to the AIM loss, we incorporate a Temporal Difference (TD) loss to propagate the proxy Q-value to the agent-collected data during self-exploration (Peng et al., 2024). This propagation generalizes our proxy Q-function $Q_\theta^I$ to the states where humans have not been intervened yet. We define the TD loss function as

$$J^{\text{TD}}(\theta) = \mathop{\mathbb{E}}_{(s,a,s')\sim\mathcal{B}_h\cup\mathcal{B}_r}\left[\left|Q_\theta(s,a)-\gamma\max_{a'}Q_{\hat{\theta}}(s',a')\right|^2\right], \tag{4}$$

where $Q_{\hat{\theta}}$ represents a target network with update delay. Integrating the TD loss allows AIM to anticipate potential agent mistakes in the future and request human assistance before they occur, while prior uncertainty-based approaches only estimate the action variance at the current state. The final loss of AIM is as follows:

$$J(\theta) = J^{\text{AIM}}(\theta) + J^{\text{TD}}(\theta). \tag{5}$$

### 4.3. Algorithm

We outline the workflow of the Adaptive Intervention Mechanism (AIM) in Alg. 2. First, we let the human expert inter-

---

**Algorithm 2** Adaptive Intervention Mechanism (AIM)

**Input:** Hyperparameter $\delta$.
Run human-gated IIL (Alg. 1) for $n$ trajectories.
Initialize the proxy Q-function $Q_\theta^I$ with $J(\theta)$ in Eq. 5.
Initialize the switch-to-human threshold $\beta$ by Eq. 6.
Initialize the switch-to-agent threshold $\epsilon$ by Eq. 8.
**for** trajectory $i = 1, 2, \dots$ **do**
  **for** timestep $t = 1, 2, \dots$ **do**
    Agent samples action $a_r \sim \pi_r(s_t)$.
    **if** $Q_\theta^I(s,a_r) > \beta$ **then**
      Human takes action $a_h \sim \pi_h(s_t)$.
      **repeat**
        Observe $s_{t+1} \sim \mathcal{P}(\cdot \mid s_t, a_h)$.
        Add $(s_t, a_h, s_{t+1})$ to the human buffer $\mathcal{B}_h$.
        Train $\pi_r$ on $\mathcal{B}_h$ with the loss function Eq. 9.
        $t \leftarrow t+1$.
        Agent samples action $a_r \sim \pi_r(s_t)$.
        Human takes action $a_h \sim \pi_h(s_t)$.
      **until** $\|a_r - a_h\|^2 \le \epsilon$.
    **end if**
    Observe $s_{t+1} \sim \mathcal{P}(\cdot \mid s_t, a_r)$.
    Add $(s_t, a_r, s_{t+1})$ to the novice buffer $\mathcal{B}_r$.
    Train $Q_\theta^I$ with the loss function $J(\theta)$ in Eq. 5.
    Update the switch-to-human threshold $\beta$ by Eq. 6.
  **end for**
**end for**
**Output:** Policy $\pi_r$ and proxy Q-function $Q_\theta^I$.

---

act with the agent in the first $n$ trajectories and provide corrective demonstrations through active interventions following Alg. 1. The human demonstrations $\mathcal{B}_h = \{(s, a_h, s')\}$ and the agent's self-exploration data $\mathcal{B}_r = \{(s, a_r, s')\}$ collected in this process are used to train an initial proxy Q-value $Q_\theta^I$ for robot-gated intervention. Then, we need to set a switch-to-human threshold $\beta$ such that the agent requests humans' help if $Q_\theta^I(s, a_r) > \beta$. We set a hyperparameter $\delta \in (0, 1)$ and define $\beta$ as the $(1-\delta)$-th quantile of $Q_\theta^I(s, a_r)$ where $s$ is sampled from the novice buffer $\mathcal{B}_r$ and $a_r \sim \pi_r(s)$. We denote

$$\beta = \mathop{\text{quantile}}_{s\sim\mathcal{B}_r, a_r\sim\pi_r(s)}(Q_\theta^I(s,a_r), 1-\delta), \tag{6}$$

$$I^r(s,a_r) = \mathbb{I}[Q_\theta^I(s,a_r) > \beta], \tag{7}$$

where $I^r(s, a_r)$ is the switch-to-human function. Following prior works on robot-gated IIL (Hoque et al., 2021a;b), when the expert is currently intervening at state $s$, the agent stops requesting human intervention from the next step if $\|a_r - a_h\|^2 \le \epsilon$, where

$$\epsilon = \mathop{\mathbb{E}}_{(s,a_h)\sim\mathcal{B}_h, a_r\sim\pi_r(s)}\left[\|a_r - a_h\|^2\right]. \tag{8}$$

We set the continue-with-human function as $I^h(s, a_r, a_h) = \mathbb{I}[\|a_r - a_h\|^2 > \epsilon]$ following Eq. 2.

Then, we start the robot-gated interactive imitation learning process. The agent requests human assistance if $I^r(s, a_r) = 1$, and resumes autonomous control if $I^h(s, a_r, a_h) = 0$. After each human demonstration, AIM updates the switch-to-human threshold $\beta$ following Eq. 6, the agent policy $\pi_r$, and the proxy Q-function $Q_\theta^I$ following Eq. 5.

Note that AIM is compatible with discrete action spaces, as we can set the switch-to-human function as $f(a_r, a_h) = \mathbb{I}[a_r \neq a_h]$. Compared with human-gated IIL methods, AIM only requires humans to proactively provide human-gated corrective demonstrations in the initial one or two trajectories ($n \leq 2$), significantly reducing the cognitive load of human in later training. In addition, in AIM and all the baselines of interactive imitation learning, we train the agent policy $\pi_r$ by imitation learning on $\mathcal{B}_h$ as in HG-DAgger (Kelly et al., 2019) to compare their intervention criteria fairly, and the loss function of $\pi_r$ follows

$$\mathcal{L}(\pi_r) = \mathbb{E}_{(s,a_h)\sim\mathcal{B}_h} \left[ \|\pi_r(s) - a_h\|^2 \right]. \quad (9)$$

## 5. Experiments

In this section, we conduct experiments to answer the following questions: (1) Does our algorithm require fewer expert demonstrations and efforts to learn a near-optimal policy than other interactive imitation learning methods? (2) Does the learned intervention criterion help the agent receive sufficient human guidance at safety-critical states, thereby capturing all necessary information for effectively imitating the expert? To investigate these questions, we conduct experiments on various reinforcement learning tasks with different state and action spaces.

### 5.1. Tasks

We consider the MetaDrive driving experiments (Li et al., 2022a) with continuous action spaces and MiniGrid Four Room task (Chevalier-Boisvert et al., 2018) with discrete action spaces. In MetaDrive, the agent needs to navigate towards the destination in heavy-traffic scenes without crashing into obstacles and other vehicles. The agent uses the sensory state vector $s \in \mathbb{R}^{259}$ as its observation and outputs a control signal $a = (a_0, a_1) \in [-1, 1]^2$ representing the steering angle and the acceleration, respectively. We evaluate the agent's learned policy in a held-out test environment separate from the training environments. In MiniGrid, the agent observes its local neighborhood and learns a navigation policy to open each door on its way to the goal.

Experiments involving real human participants are time-consuming, and their interaction results vary significantly in different trials. Following the prior works on interactive imitation learning (Hejna et al., 2023; Peng et al., 2021), we incorporate well-trained neural policies in the training loop to approximate human policies. In the initial $n$ trajectories

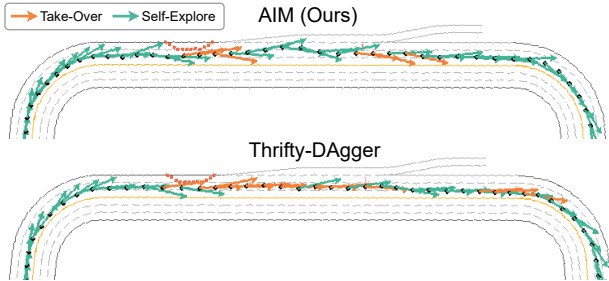

*Figure 3.* Comparison of intervention strategies between AIM and Thrifty-DAgger in MetaDrive after 2K training steps. Arrows indicate steering actions in a top-down view. AIM only requests expert help near traffic cones and roadblocks, whereas Thrifty-DAgger frequently requests takeovers even on straight roads. See Appendix A for details.

that require human-gated interventions, the neural expert follows Eq. 2 to provide corrective demonstrations if the action-difference function $f(a_r, a_h) = 1$. During the evaluation, we report the average of all the metrics obtained by the learned agent alone in 50 rollouts without interaction with the neural experts.

### 5.2. Experimental Setting

**Evaluation Metrics** We define the *expert-involved steps* as the total environment steps that require the expert to monitor, a measure of the expert's effort in guiding the agent during training. We limit the total expert-involved steps to 2000 and report the total number of expert-involved transitions (*expert data usage*) and the *overall intervention rate*, which is the ratio of expert data usage to the total data usage. In MiniGrid, we report the *success rate* as the agents' evaluation metric. The success rate is the ratio of episodes where the agent reaches the goal to the total evaluation rollouts. In MetaDrive, we also report the *episodic return* and *route completion rate* during evaluation. The route completion rate is the ratio of the agent's successfully traveled distance to the length of the complete route.

**Experimental Details** The neural expert is trained using PPO-Lagrangian (Ray et al., 2019) with 20M environment steps. We train each interactive imitation learning baseline five times using distinct random seeds. Then, we roll out 50 trajectories generated by each model in the held-out evaluation environment and average each evaluation metric as the model's performance. We report the average performance of the best checkpoints from the five random seeds as the result of each baseline. We also provide the standard deviation of each metric among the five runs of each baseline. In AIM, we set $\delta = 0.05$. For fairness, we introduce the same warm-up stage to AIM and each baseline by letting the expert monitor the initial $n = 2$ trajectories.

*Table 1.* Comparison of methods with training/testing statistics in the MetaDrive environment with 2000 expert-involved steps. The overall intervention rate is given besides the expert data usage. We report the average performance and standard deviation of the best checkpoints from five random seeds.

| Method | Robot-Gated | Training | | Testing | | |
|---|---|---|---|---|---|---|
| | | Expert Data Usage | Total Data Usage | Success Rate | Episodic Return | Route Completion |
| Neural Expert | – | – | – | $0.84_{\pm 0.05}$ | $336.5_{\pm 17.1}$ | $0.93_{\pm 0.01}$ |
| BC | – | 2K | 2K | $0.33_{\pm 0.04}$ | $243.0_{\pm 46.7}$ | $0.62_{\pm 0.08}$ |
| HG-DAgger | ✗ | 0.9K (0.45) | 2K | $0.61_{\pm 0.07}$ | $310.8_{\pm 16.7}$ | $0.78_{\pm 0.07}$ |
| PVP | ✗ | 0.4K (0.19) | 2K | $0.62_{\pm 0.06}$ | $270.4_{\pm 28.6}$ | $0.77_{\pm 0.04}$ |
| Ensemble-DAgger | ✓ | 2K (0.55) | 3.6K | $0.60_{\pm 0.09}$ | $267.4_{\pm 9.9}$ | $0.54_{\pm 0.10}$ |
| Thrifty-DAgger | ✓ | 2K (0.21) | 9.5K | $0.58_{\pm 0.03}$ | $250.0_{\pm 23.9}$ | $0.73_{\pm 0.03}$ |
| AIM (Ours) | ✓ | 1.9K (0.24) | 7.7K | $\mathbf{0.82}_{\pm 0.06}$ | $\mathbf{328.4}_{\pm 20.4}$ | $\mathbf{0.91}_{\pm 0.03}$ |

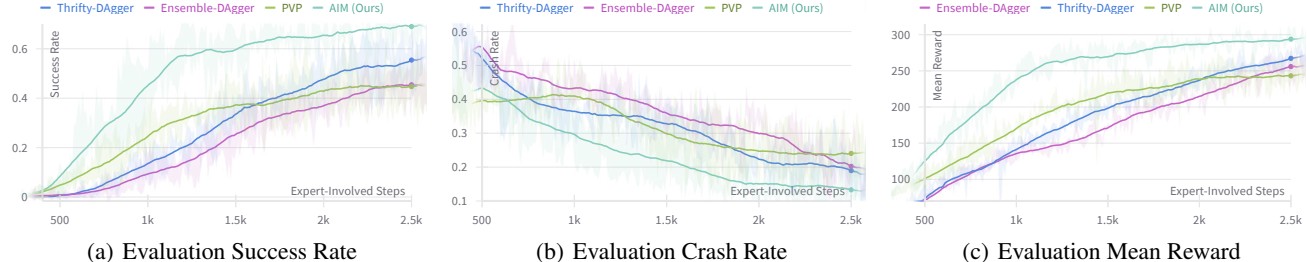

(a) Evaluation Success Rate      (b) Evaluation Crash Rate      (c) Evaluation Mean Reward

*Figure 4.* Training curves of Interactive Imitation Learning (IIL) approaches in MetaDrive. We report the evaluation metrics of the corresponding policy in a held-out test environment, where evaluation is conducted after expert-in-the-loop training without expert involvement. We apply a running average with a window length of 40 to smoothen the curves. AIM imitates the expert's behavior with fewer expert-involved steps than all the baselines, achieving a higher success rate and episodic return, as well as a lower safety cost.

**Baselines** We test these robot-gated interactive imitation learning baselines: Behavioral Cloning (BC), Ensemble-DAgger (Menda et al., 2019), and Thrifty-DAgger (Hoque et al., 2021a). We also evaluate the human-gated interactive imitation learning methods: Human-Gated DAgger (HG-DAgger) (Kelly et al., 2019) and Proxy Value Propagation (PVP) (Peng et al., 2024). Fig. 4 plots each method's evaluation metrics against the number of expert-involved steps. For the human-gated PVP baseline, the expert-involved steps correspond to the total data usage because the expert monitors the entire training process. For AIM and the robot-gated baselines, the expert-involved steps correspond to the expert data usage. We compare the performance of each method under the same 2K expert-involved-step budget in Table 1 and Table 2.

**5.3. Baseline Comparison**

In Table 1, we compare the performance of our proposed AIM algorithm against several baseline methods in the MetaDrive simulator (Li et al., 2022a). Our algorithm AIM achieves a higher success rate and episodic return compared with all the robot-gated IIL baselines under the same amount of expert demonstrations. Across both robot-gated and human-gated IIL baselines, AIM outperforms all com-

petitors while markedly reducing expert-involved steps (see Fig. 4), thereby easing the expert's monitoring effort.

We also compare the performance of AIM with other IIL baselines in the MiniGrid Four Room task (Chevalier-Boisvert et al., 2018) with discrete action spaces. From Table 2, we can conclude that AIM also outperforms these baselines and alleviates the burdens on human supervisors in environments with discrete action spaces.

**5.4. A Case Study in a Toy MetaDrive Environment**

Fig. 3 visualizes the intervention strategies of AIM and Thrifty-DAgger after 2K training steps in a simplified MetaDrive setting. The toy scenario consists of a straight road flanked by a set of traffic cones and a single roadblock.

From Fig. 3, AIM requests expert help exclusively near the traffic cones and the roadblock while remaining autonomous in the unobstructed areas. Such spatially focused queries reflect an efficient intervention strategy: the agent solicits demonstrations precisely where data are scarce and errors are most likely. In contrast, Thrifty-DAgger issues queries along the entire obstacle corridor, including many states the agent has already encountered frequently, thereby involving unnecessary expert effort.

*Table 2.* Comparison of methods with training/testing statistics in the Minigrid Four Room environment with no more than 2000 expert-involved steps. The overall intervention rate is given besides the expert data usage.

| Method | Robot-Gated | Training | | Testing |
|---|---|---|---|---|
| | | **Expert Data Usage** | **Total Data Usage** | **Success Rate** |
| Neural Expert | – | – | – | $0.78 \pm 0.03$ |
| BC | – | 2K | 2K | $0.01 \pm 0.0$ |
| HG-DAgger | ✗ | 0.2K (0.12) | 2K | $0.20 \pm 0.04$ |
| PVP | ✗ | 0.2K (0.12) | 2K | $0.34 \pm 0.10$ |
| Ensemble-DAgger | ✓ | 2K (0.36) | 5.6K | $0.38 \pm 0.08$ |
| Thrifty-DAgger | ✓ | 2K (0.27) | 7.4K | $0.42 \pm 0.12$ |
| AIM (Ours) | ✓ | 0.4K (0.09) | 4.4K | $\mathbf{0.63} \pm \mathbf{0.05}$ |

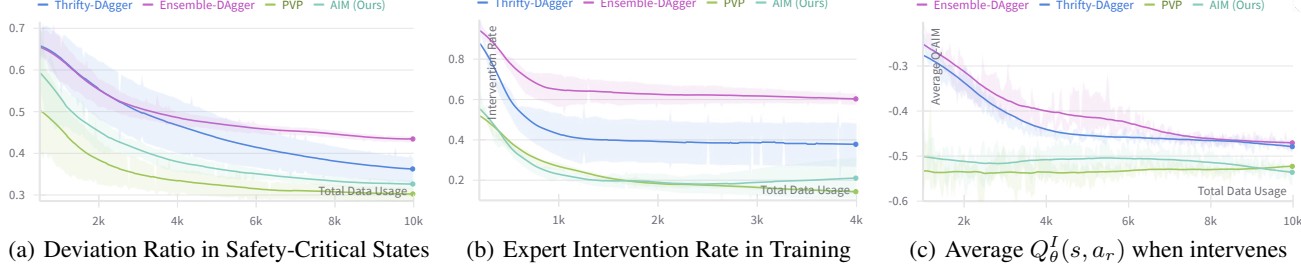

(a) Deviation Ratio in Safety-Critical States     (b) Expert Intervention Rate in Training     (c) Average $Q_\theta^I(s, a_r)$ when intervenes

*Figure 5.* Training statistics of IIL approaches in MetaDrive (lower is better). (a) We define the deviation ratio as the fraction of safety-critical states in which the agent's action deviates from the expert's. AIM drives a faster decline than Ensemble-DAgger and Thrifty-DAgger, matching the human-gated PVP. (b) AIM and PVP require a comparable number of expert demonstrations, and AIM's intervention frequency stays below Ensemble-DAgger and Thrifty-DAgger during the first 4K steps. (c) AIM aligns agent actions with human behavior more rapidly than Ensemble-DAgger and Thrifty-DAgger.

Fig. 8 extends this comparison across multiple training stages. It shows that AIM progressively reduces its query frequency as the agent masters each corner case, whereas Thrifty-DAgger continues to request assistance repeatedly in the same safety-critical regions.

### 5.5. Effectiveness of Our Intervention Strategy

In this section, we address the second question: Does the learned intervention criterion help the agent receive sufficient human guidance at safety-critical states, thereby capturing all necessary information for effectively imitating the expert?

To better investigate this problem, we define a safety-critical state $s$ in MetaDrive as one where the PPO expert's action $a_h = (a_h^{\text{steer}}, a_h^{\text{throttle}})$ with its 2-norm $\|a_h\|_2 > 0.5$, i.e., a sudden brake or a sharp turn. Recall that we are working with a PPO-Lagrangian expert therefore we can query its actions at all states, even though the intervention might not be active. Within these safety-critical states, we label the state where the agent action deviates from the expert, i.e. $f(a_r, a_h) = 1$, as a deviation state. In Fig. 5(a), we plot the ratio between the number of deviation states and the safety-critical states. Compared with other robot-gated IIL baselines Ensemble-DAgger (Menda et al., 2019)

and Thrifty-DAgger (Hoque et al., 2021a), AIM sustains a markedly lower deviation ratio throughout training, showing that it aligns with the expert far more quickly, especially in truly hazardous situations where a sudden turn or an immediate brake is required. This trend reveals that fixed-threshold uncertainty heuristics under-query the expert when safety is at risk, allowing errors to persist and wasting environment samples. By contrast, the proxy Q-function in AIM triggers assistance precisely when needed, producing a sharper decline in the deviation ratio. Furthermore, after 5K environment steps, the AIM and PVP curves nearly coincide, indicating that AIM attains the expert-level behavior of the human-gated method without requiring continuous human monitoring. In Fig. 5(b) we present the overall expert intervention rate, complementing the deviation analysis above. Surprisingly, AIM attains its superior alignment with the expert while requesting fewer interventions than both Ensemble-DAgger and Thrifty-DAgger.

In addition, we compare how quickly each baseline aligns the agent's actions with expert intent in Fig. 5(c). To this end, we train an AIM proxy Q-function $Q_\theta^I$ using Eq. 5 *for each baseline* to measure how much a state-action pair $(s, a_r)$ deviates from expert behavior. We recall that a high Q-value implies that the agent's action violates expert intent. Fig. 5(c) indicates that Ensemble-DAgger and Thrifty-

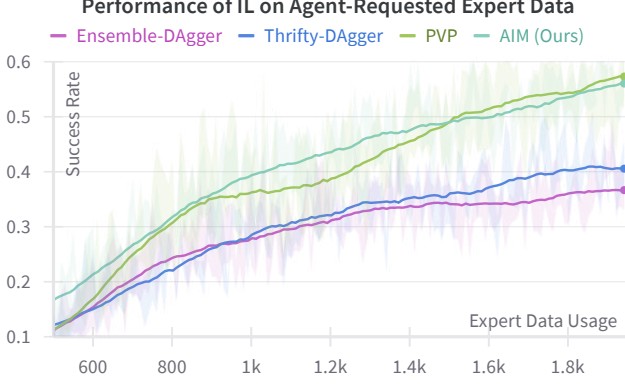

**Figure 6.** We plot the success rates of the agent trained by Behavioral Cloning using the human buffer collected at different training stages. This offline-trained agent's performance reflects the quality of the expert data collected by each method. The near overlap between the AIM and human-gated PVP curves shows that the demonstrations AIM requests are at least as informative as those gathered under continuous human supervision.

DAgger require more environment data before their agents' actions conform to expert behavior. The three subfigures in Fig. 5 also demonstrate that AIM and human-gated PVP share similar takeover mechanisms, but we note that PVP achieves it at the cost of continuous expert oversight.

In the next experiment, we investigate if AIM requests fewer human demonstrations to capture all the information needed for successful imitation. We will show that AIM-requested expert data contains rich expert guidance information in critical states, allowing us to train a new agent policy from scratch by imitation learning on this offline dataset without any environmental interaction. The experiment works as follows. After training AIM, we extract the human replay buffer $\mathcal{B}_h$. We denote $\mathcal{B}_h^T$ as the first $T$ state-action pairs in $\mathcal{B}_h$, where $T \leq 2000$. Then, we discard all the expert demonstrations collected in the initial warm-up stage from each $\mathcal{B}_h^T$, so that $\mathcal{B}_h^T$ only contains the expert demonstrations requested by the agent's intervention criterion. We utilize the dataset $\mathcal{B}_h^T$ to train an agent policy $\pi_r^T$ from scratch using imitation learning with the loss function Eq. 9. We then plot the success rate of $\pi_r^T$ w.r.t. $T$ in Fig. 6. The results show that the human replay buffer $\mathcal{B}_h$ obtained through AIM contains sufficient information needed to train a new offline IL policy. Moreover, the offline performance curve of AIM nearly coincides with that of the human-gated PVP baseline, demonstrating that the demonstrations AIM requests are at least as informative as those collected under continuous human supervision. This explains why AIM enables the agent to imitate human behavior and recovers the expert policy more rapidly and efficiently in Table 1 and Table 2.

These results indicate that AIM reduces the human demonstrations needed and enhances the effectiveness of human

**Table 3.** Ablation studies of AIM with testing statistics in the MetaDrive environment with 2000 expert-involved steps.

| Method | Testing | | |
|---|---|---|---|
| | Success Rate | Episodic Return | Route Completion |
| AIM - reward | 0.61 | 275.1 | 0.82 |
| AIM - no TD loss | 0.57 | 246.7 | 0.75 |
| **AIM (Ours)** | **0.82** | **328.4** | **0.91** |

interventions. By directing queries to states with the greatest agent-expert divergence, AIM ensures that each requested demonstration delivers high informational value, thereby facilitating more efficient policy learning without the need for continuous expert monitoring.

### 5.6. Ablation Studies of AIM

In Table 3, we perform the ablation studies of AIM in the MetaDrive environment. "AIM - reward" labels the reward function $r(s, a_h)$ as $+1$ and $r(s, a_r)$ as $-1$ at any intervention instead of labeling the $Q$ function. "AIM - w.o. TD loss" disables TD learning by setting $J^{\text{TD}}(\theta) = 0$, so the proxy Q-function is trained solely with the supervised labels from the human buffer $\mathcal{B}_h$. Table 3 shows that dropping the TD loss or replacing the Q-labeling with reward-labeling hurts the performance of AIM.

## 6. Conclusion

In this work, we propose AIM, a novel robot-gated interactive imitation learning algorithm that learns an adaptive criterion on requesting human help. By training a proxy Q-function to approximate the human-gated intervention strategy, AIM proactively requests human assistance when the agent's policy deviates from the expert. This design enables the robot-gated criterion to naturally evolve with the agent's policy, steadily reducing human interventions as the agent becomes proficient. A key factor in AIM's success is its ability to emulate the human intervention mechanism by targeting safety-critical states. By effectively capturing the most informative expert guidance information, AIM effectively reduces the expert's cognitive efforts to teach the agent in these states. Compared to existing Interactive Imitation Learning methods, AIM achieves near-optimal success rates with fewer interventions from the human supervisors, outperforming both human-gated and robot-gated baselines across MetaDrive and MiniGrid tasks.

**Limitations.** We assume that the expert knows the optimal control strategy and behaves correctly. Additionally, this paper does not include real-human experiments or user studies, and human demonstrations may be imperfect or faulty. Extending AIM to support human interactions with multiple agents remains unexplored.

## Impact Statement

Our AIM fosters a safer and more efficient interactive imitation learning framework by requesting expert guidance only when needed, minimizing human cognitive load, and enhancing evaluation performance. While AIM has the potential to accelerate the deployment of intelligent systems, developers should also avoid introducing expert biases and overdependence on automation.

## Acknowledgement

The project was supported by NSF grants CCF-2344955 and IIS-2339769. ZP is supported by the Amazon Fellowship via UCLA Science Hub.

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

## A. Comparison of Intervention Mechanisms in MetaDrive

To visualize the differences in the intervention mechanisms between AIM and the uncertainty-based method Thrifty-DAgger (Hoque et al., 2021a), we design a toy MetaDrive environment in Fig. 7. The toy environment consists of two curved segments (one at the start and one at the end) and a straight road between them. The straight road contains traffic cones and a single roadblock, as illustrated in Fig. 7(a) and Fig. 7(b), respectively. Fig. 7(c) provides a top-down view of the straight road. There are no other traffic participants in this toy environment. The expert is trained using PPO (Ray et al., 2019) for 20 million environment steps and can safely navigate to the destination without colliding with traffic cones or the roadblock.

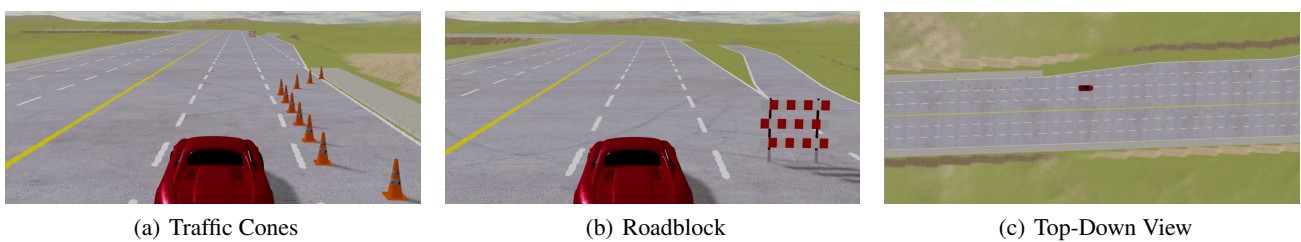

    (a) Traffic Cones          (b) Roadblock          (c) Top-Down View

*Figure 7.* Illustration of the toy MetaDrive environment. The key objective is to navigate to the destination without crashing into the traffic cones or the roadblock.

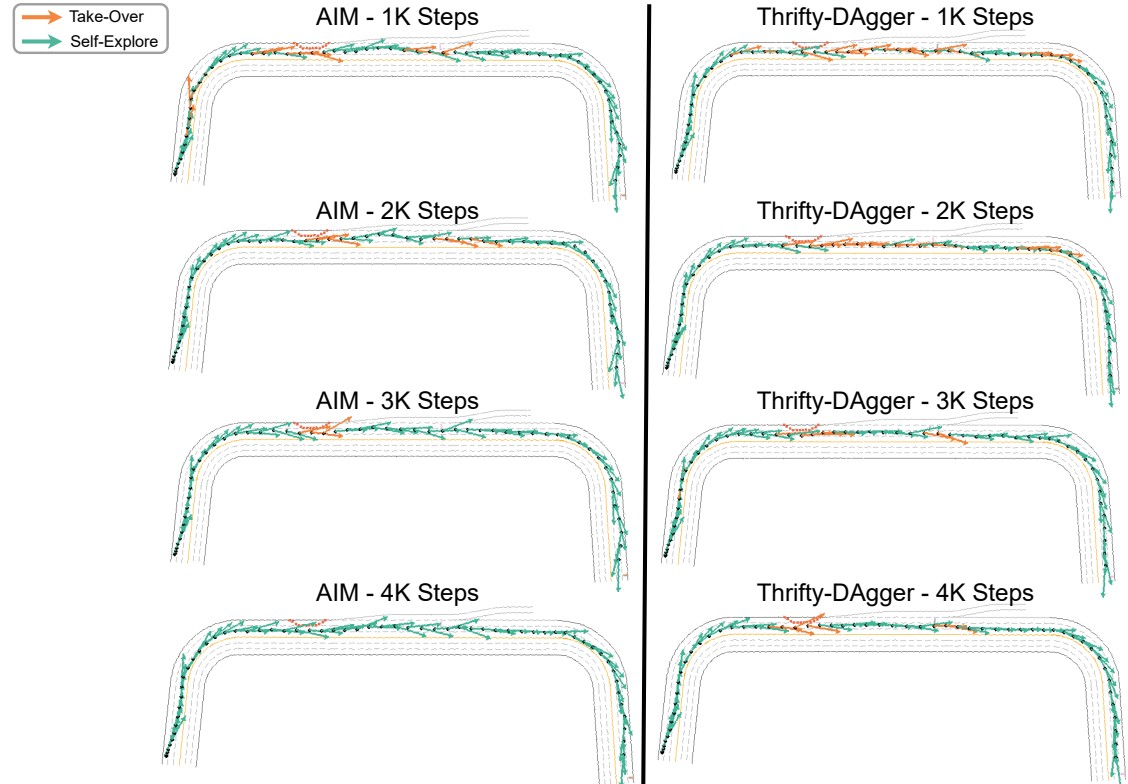

*Figure 8.* Comparison of intervention strategies between AIM and Thrifty-DAgger in MetaDrive at different training stages. Arrows indicate the agent's steering actions in a top-down view, with green arrows showing autonomous actions and yellow arrows showing expert takeovers. AIM reduces expert queries as the agent becomes more proficient, while Thrifty-DAgger continues to request expert assistance frequently on the straight road.

In Fig. 8, we compare the intervention mechanisms of AIM and Thrifty-DAgger at different training steps. After 2K steps, AIM's takeover mechanism confirms that the agent has learned to navigate curves and stay on the road, so it only requests expert help near the traffic cones and the roadblock. After 3K steps, AIM no longer requests assistance at the roadblock, and

by 4K steps, AIM agent requires no expert intervention and can confidently navigate to the goal. In contrast, Thrifty-DAgger needs to request expert help almost continuously along the straight road during the first 2K steps. Even after 4K steps, Thrifty-DAgger still requests takeovers near the traffic cones and the roadblock. This demonstrates that the uncertainty-based takeover strategy in Thrifty-DAgger fails to adapt to the agent's improving performance, thereby demanding more expert demonstrations and cognitive effort.

## B. Environment Details

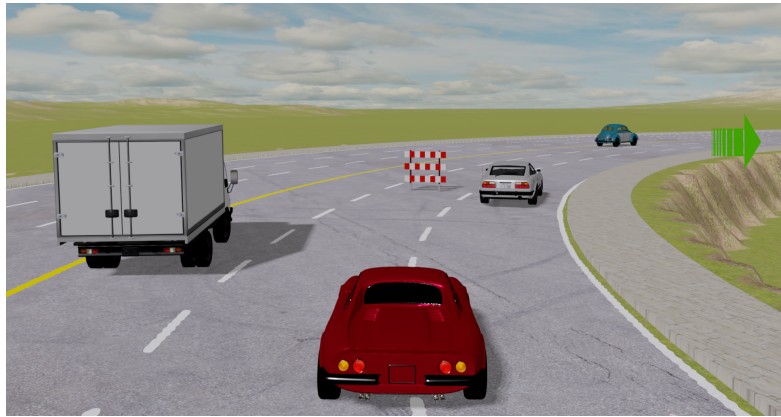

*Figure 9.* MetaDrive Safety Benchmark

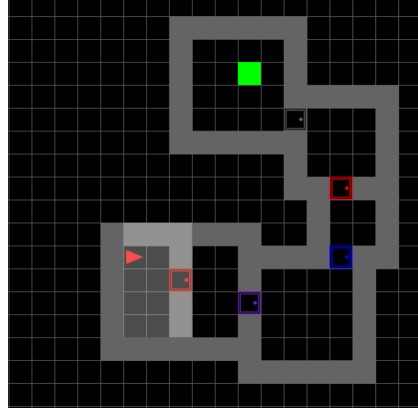

*Figure 10.* MiniGrid

**MetaDrive Safety Benchmark.** In Table 1, we compare AIM against several interactive imitation-learning baselines on the MetaDrive safety benchmark (Li et al., 2022a). In this environment, the agent controls a vehicle using low-level throttle, brake, and steering inputs to reach the goal while avoiding collisions, as shown in Fig. 9. MetaDrive's safety benchmark evaluates the agent's performance by procedurally synthesizing diverse driving scenarios: the training split consists of 50 unique maps, and the test split contains another 50 distinct maps. Each episode randomly selects a road layout and randomizes the spawn positions of both traffic and the ego vehicle, ensuring that agents encounter novel situations during both training and testing. In the MetaDrive safety benchmark, the sensory state vector is used as the agent's observation. The agent's observation includes: (1) its current state variables such as the steering angle, heading, speed, and distance to lane boundaries; (2) the navigation cues that indicate the direction toward the checkpoints and destination; and (3) the surrounding information represented by a 240-dimensional Lidar-like point cloud with a 50m detection range, capturing nearby vehicles and obstacles. We implement both the agent's policy and AIM's proxy Q-function using a two-layer MLP architecture.

**MiniGrid Multi-Room Environment.** In Table 2, we evaluate AIM in the MiniGrid task shown in Fig. 10. The MiniGrid multi-room task requires extensive exploration: the agent must navigate through a sequence of doors, opening each one in turn, before finally reaching the green goal square (Chevalier-Boisvert et al., 2018). The agent's starting position, the goal location, the positions of all doors, and the room geometries are randomized in each episode. The state and action spaces are discrete. The agent's observation is represented by the semantic map of the agent's local neighborhood, rendered as a 7×7 grid; the agent cannot see beyond walls or other obstacles. The action space has four valid actions: turn left, turn right, move forward, and open the door.

## C. Hyperparameters

In MetaDrive, we implement the control policy and the proxy Q-network using a two-layer MLP, where each hidden layer has 256 units with ReLU activations. For MiniGrid tasks, all models employ a three-layer convolutional network with filter sizes of 16, 16, and 32. Each convolution uses a 2×2 kernel, and a max-pooling layer is inserted between the first and second convolutional layers. The ReLU activation is applied after each layer. When training AIM and the robot-gated baselines Ensemble-DAgger and Thrifty-DAgger, we use the same switch-to-agent threshold following Eq. 6.

*Table 4.* AIM (MetaDrive)

| Hyper-parameter | Value |
| --- | --- |
| Discounted Factor $\gamma$ | 0.99 |
| Learning Rate | 1e-4 |
| Gradient Steps per Iteration | 1 |
| Train Batch Size | 1024 |
| Switch-to-Human Quantile $\delta$ | 0.05 |

*Table 5.* AIM (MiniGrid)

| Hyper-parameter | Value |
| --- | --- |
| Discounted Factor $\gamma$ | 0.99 |
| Learning Rate | 1e-4 |
| Gradient Steps per Iteration | 32 |
| Train Batch Size | 200 |
| Switch-to-Human Quantile $\delta$ | 0.05 |

*Table 6.* PVP (MetaDrive)

| Hyper-parameter | Value |
| --- | --- |
| Discounted Factor $\gamma$ | 0.99 |
| $\tau$ for Target Network Update | 0.005 |
| Learning Rate | 1e-4 |
| Gradient Steps per Iteration | 1 |
| Train Batch Size | 1024 |
| Free Level | 0.95 |

*Table 7.* PVP (MiniGrid)

| Hyper-parameter | Value |
| --- | --- |
| Discounted Factor $\gamma$ | 0.99 |
| $\tau$ for Target Network Update | 0.005 |
| Learning Rate | 1e-4 |
| Gradient Steps per Iteration | 32 |
| Target Network Update Interval | 1 |
| Train Batch Size | 200 |
| Free Level | 0.95 |

*Table 8.* Ensemble-DAgger (MetaDrive)

| Hyper-parameter | Value |
| --- | --- |
| Number of Instances | 5 |
| Learning Rate | 1e-4 |
| Gradient Steps per Iteration | 1 |
| Train Batch Size | 1024 |
| BC Warmup Steps | 200 |
| Switch-to-Human Threshold $\varepsilon$ | 1e-3 |

*Table 9.* Ensemble-DAgger (MiniGrid)

| Hyper-parameter | Value |
| --- | --- |
| Number of Instances | 5 |
| Learning Rate | 1e-4 |
| Gradient Steps per Iteration | 32 |
| Train Batch Size | 200 |
| BC Warmup Steps | 100 |
| Switch-to-Human Threshold $\varepsilon$ | 5e-3 |

*Table 10.* Thrifty-DAgger (MetaDrive)

| Hyper-parameter | Value |
| --- | --- |
| Number of Instances | 5 |
| Discounted Factor $\gamma$ | 0.99 |
| Learning Rate | 1e-4 |
| Gradient Steps per Iteration | 1 |
| Train Batch Size | 1024 |
| BC Warmup Steps | 200 |
| Switch-to-Human Quantile of Novelty $\delta_1$ | 0.05 |
| Update Frequency of $\delta_1$ | 25 |
| Switch-to-Human Quantile of Risk $\delta_2$ | 0.01 |
| Update Frequency of $\delta_2$ | 25 |

*Table 11.* Thrifty-DAgger (MiniGrid)

| Hyper-parameter | Value |
| --- | --- |
| Number of Instances | 5 |
| Discounted Factor $\gamma$ | 0.99 |
| Learning Rate | 1e-4 |
| Gradient Steps per Iteration | 32 |
| Train Batch Size | 200 |
| BC Warmup Steps | 100 |
| Switch-to-Human Quantile of Novelty $\delta_1$ | 0.05 |
| Update Frequency of $\delta_1$ | 25 |
| Switch-to-Human Quantile of Risk $\delta_2$ | 0.01 |
| Update Frequency of $\delta_2$ | 25 |

