# OpenReview forum: "Robot-Gated Interactive Imitation Learning with Adaptive Intervention Mechanism"
_ICML.cc/2025/Conference — ICML 2025 poster_

### Official Review · Reviewer_JBzX · 2025-03-14

**Overall Recommendation:** 3

**Summary:**

The paper proposes an adaptive intervention strategy aiming to use the shared automony to improve the robot execution process. Previous robot -gated designs rely on the entropy to judge whether to let the human intervent. Using this strategy, the robot would frequently ask humans for the help, whihch is constly. Authors propose an a adaptie interactive intervention straetey, where Q function is learned, whch is further used to judge when to interact. Experiments demonstrated the effectiveness enabled by the method.

**Claims And Evidence:**

The claims made  are supported by clear and convincing evidence.

**Essential References Not Discussed:**

References are properly discussed.

**Experimental Designs Or Analyses:**

Experiemtnal designs are reasonalbe but a little bit simple.

The experiments mainly focus on evaluating the performance evolution of the agent as the number of human-involved steps increases. By comparing with previous robot-gated and human-gated imitation learning methods, authors wish to demonstrate that their method can achieve higher performance in a shorter time with fewer human-invovled steps. They also demonstrate that their method can help the agent receive sufficient human guidance at safety-critical issues.

The effectiveness of the proposed method can be validated by quantitative results shown in the table and plots. The main limitation lies in the simplicity of their evaluated tasks. Besides, qualitative evaluations are missing.

**Methods And Evaluation Criteria:**

The method and evaluation criteria make sense for the problem.

**Other Comments Or Suggestions:**

N/A

**Other Strengths And Weaknesses:**

Strengths:
- The motivating example in the method section is a good point and can help readers understand the problem setting and motivations.

Weaknesses:
- Line 053: "so the agent keeps requesting help at a fixed rate even when it has..." why "fixed intervention criterion" would cause "fixed request rate"? The logic between such two sentences is not clear.
- Line 071: "Second, our learned intervention criterion dynamically adjusts to the frequently changing agent policy during training." The meaning of this sentence is also unclear. What do you mean by  "dynamically adjusts to...agent policy"? That's quite confusing.
- Line 295: "evaluation performance"? That's strange. "we report the success rate" or "we employ the success rate as the evaluation metric" would be better

**Questions For Authors:**

- Effecitveness in more complex tasks: Could the authors demonstrate the value of their method in more complex tasks such as those involving physical environments, e.g. Mojoco tasks like Ant in Gymnasium.
- Method design: why the proposed Q function could help the agent receive sufficient human guidance at "safety-critical states"? Why the Q-function can emphasize the "safety-critical states"?

**Relation To Broader Scientific Literature:**

Related to the robotics and adaptive policies.

**Theoretical Claims:**

N/A

---

> ### Author Rebuttal · Authors · 2025-03-31
>
> Thank you for your effort to thoroughly review our paper and for your feedback. In response to your feedback, we have included qualitative evaluations and ablation studies that have strengthened the study.
>
> __Experimental Designs Or Analyses:__
>
> >1.Qualitative evaluations are missing.
>
> We include qualitative evaluations in Fig 1 of https://limewire.com/d/teXVP#Okg7PsYIne to show that when the car approaches the road boundaries, AIM successfully requests human intervention, while other uncertainty-based methods output a low uncertainty estimate and fail to signal the need for help.
>
> __Other Strengths And Weaknesses:__
>
> >1.Line 053: why "fixed intervention criterion" would cause "fixed request rate"? The logic between such two sentences is not clear.
>
> In Line 053, __fixed intervention criterion__ refers to requesting human help when the __uncertainty estimate exceeds a constant $\epsilon$__, as in EnsembleDAgger, which uses the variance of actions in the policy ensemble as the uncertainty estimate.
>
> According to Fig 4 of https://limewire.com/d/teXVP#Okg7PsYIne, __EnsembleDAgger requests expert help at a fixed intervention rate__ after 1K steps, implying that some states’ uncertainty estimates always exceed $\epsilon$. In contrast, ThriftyDAgger adapts $\epsilon$ and avoids a fixed request rate.
>
> Therefore, we __revise Line 053__ as follows:
>
> "The uncertainty-based methods including (Menda et al., 2019; Kelly et al., 2019) request human help when __the uncertainty estimate is larger than a fixed threshold__. __Without adjusting the threshold adaptively__, the agent keeps requesting expert help even when it has successfully imitated the human expert."
>
> >2.Line 071: What do you mean by "dynamically adjusts to...agent policy"? That's quite confusing.
>
> We __revise Line 071__ for clarity:
>
> "Our learned intervention criterion adaptively __shrinks the intervention rate as the agent becomes performing__. Our learned intervention criterion can request human help at an appropriate rate based on the agent's performance during training."
>
> We further explain Line 071 as follows. According to Line 184-188, the __intervention frequency of AIM gradually drops down__ as the agent’s performance improves during training. The reason is that AIM’s Q value depends on the proportion of states that the agent’s action aligns with the expert’s action. During training, the average of AIM’s Q value decreases towards -1, leading to the shrink of the intervention rate. The shrink of the intervention rate is also shown in Fig 4 of https://limewire.com/d/teXVP#Okg7PsYIne . We can observe that our method’s intervention rate matches that of human-gated PVP even though it’s a __robot-gated__ method, implying that our intervention mechanism adapts to the agent’s performance and resembles the human-gated intervention rule. In addition, our intervention rate is lower than other robot-gated baselines.
>
> >3.Line 295: "we report the success rate" or "we employ the success rate as the evaluation metric" would be better.
>
> Thanks for your suggestion! We revise Line 295 as follows:
> "In MiniGrid, we employ the success rate as the agent’s evaluation metric."
>
> __Questions For Authors:__
>
> >1. Could the authors demonstrate the value of their method in more complex tasks such as those involving physical environments, e.g. Mujoco tasks like Ant in Gymnasium.
>
> We appreciate the reviewer’s interest in evaluating our method on physical tasks such as those in MuJoCo. However, tuning and training a near-optimal expert and all baseline methods demand time and resources. Moreover, human demonstrations for MuJoCo tasks like Ant are infeasible, as a human cannot reasonably demonstrate the complex multi-legged locomotion task. Our MetaDrive environment with a high-dimensional observation space generates diverse driving scenarios such as fixed or movable traffic vehicles, traffic cones, and warning triangles, which already offers a compelling and challenging benchmark. We plan to include experiments on physical tasks in future work.
>
> >2. Why the proposed Q function could help the agent receive sufficient human guidance at "safety-critical states"? Why the Q-function can emphasize the "safety-critical states"?
>
> The key is that the proposed Q-function can classify states where __the agent’s actions align with human actions__ and those where __there is a significant action discrepancy__.
>
> By explicitly using an action difference function $f$ to label these states, the Q-function learns to emphasize those states where human guidance is most needed. Additionally, incorporating a TD loss propagates these signals to nearby states.
>
> We also include an ablation study in Fig 3 of https://limewire.com/d/teXVP#Okg7PsYIne , showing that dropping the TD loss or replacing the Q-labeling by reward-labeling will damage the performance of AIM. This implies the effectiveness of our AIM Q-function design.

---

### Official Review · Reviewer_xbPG · 2025-03-15

**Overall Recommendation:** 3

**Summary:**

This paper develops an approach to imitation learning called Adaptive Intervention Mechanism (AIM) that learns whether to ask an expert for an action label based upon whether AIM thinks the imitation learner already knows the correct action. An objective function is developed (Equation 3) that governs this AIM scheme by learning the weights of a Q-function, weighted by the level of disagreement between the expert and learner. Results are generated on some simple benchmarks and show that the proposed approach generally outperforms some baselines.

### Update after rebuttal

The reviewer appreciates the authors’ response and responded below. Overall, the response from the authors suggests a plan to improve the paper that would indeed improve the contribution to ICML. However, it would have been helpful to actually see these changes and have more details to ensure that the paper's claims match its contribtuions. I was initially a weak accept, and the rebuttal has solidified that rating for me.

**Claims And Evidence:**

The claims that AIM is a novel algorithm, that the algorithm was evaluated and showed positive results is generally accurate. However, the term "sufficient" in the third claim (Line 98) should be softened unless providing a proof.

**Essential References Not Discussed:**

The paper does not discuss Confidence-based Autonomy (CBA).

Chernova, S. and Veloso, M., 2009. Interactive policy learning through confidence-based autonomy. Journal of Artificial Intelligence Research, 34, pp.1-25.

This paper sets up essentially the same problem and an analogous solution approach. There is an imitation learner that decides when to ask humans to take over control (add labels) vs. to autonomous execute (which still allows humans to observe and manually override). While Chernova's work was based upon a measure of model uncertainty (confidence), it is still directly related to how this paper uses a Q-function (albeit one I am confused about -- see my question regarding whether a TD-error actually means anything or exists). At a minimum, this prior work must be discussed, and the authors should do a more thorough literature review to find related papers that they might have missed by focusing only on recent trends in gated DAgger-like approaches. Ideally, this paper would benchmark against CBA.

**Experimental Designs Or Analyses:**

The experimental design has a key weakness that it lacks a user study. Some of the results appear mixed (e.g., Figures 4-5). However, the analysis is reasonable for an ICML paper.

**Methods And Evaluation Criteria:**

Yes, generally speaking, the approach and how it is evaluated make sense. However, a user study would have been helpful. There are issues described below.

**Other Comments Or Suggestions:**

There is a missing article in "mimic human intervention rule"

**Other Strengths And Weaknesses:**

Imitation Learning is, by definition [Ross et al., 2011], interactive and online. Humans give labels in real-time based upon online policy rollouts. The notion of "interactive imitation learning" seems to be confusing established concepts. The paper does provide references (such as Kelly et al., 2019) to back up this terminology, but I think it is unhelpful. If this paper refers to "imitation learning" in the sense of any learning from demonstration algorithm (whether it be Behavior Cloning or Inverse Reinforcement Learning (IRL)), then that needs to be clearer. Also, IRL is non-interactive but online. There is also offline IRL, etc. I think a more wholistic view of the literature and reaching further back into established concepts would improve the paper. I'd also refer the authors to this book [Chernova & Thomaz, 2022], which is a helpful treatise to define these concepts.

Ross, S., Gordon, G. and Bagnell, D., 2011, June. A reduction of imitation learning and structured prediction to no-regret online learning. In Proceedings of the fourteenth international conference on artificial intelligence and statistics (pp. 627-635). JMLR Workshop and Conference Proceedings.

Chernova, S. and Thomaz, A.L., 2022. Robot learning from human teachers. Springer Nature.

As per Lines 275-278, the paper says, "Following the prior works on interactive imitation learning (Hejna et al., 2023; Peng et al., 2021), we incorporate well-trained neural policies in the training loop to approximate human policies."

However, the limitations section is quite short and does not fully address the many weaknesses of not doing a real user study. Users are not "perfect," so claims should be softened.

Figures 4-5 show unconvincing results regarding the superiority of AIM vs. PVP.

It would have been helpful to include Adversarial Inverse Reinforcement Learning or a more competitive baseline. AIM requires online interaction with the user and the environment. AIRL may do quite well if given access to the environment but only limited data.

**Questions For Authors:**

Under what conditions of f in Equation 2 is the agent able to recover the expert policy? Must it conform to the equation embedded in the text from Menda et al., 2019 and Hoque et al., 2021a;b? This claim seems to strong without offering further proof, and the description should be clearer if the authors are relying on prior work for the proof.

It is confusing to say that $Q^I_{\theta}(s, a_r)$ is equal to -1 or +1 by assignment (Lines 209-218). Should it not be the reward function definition for -1 and +1 -- not the Q-function? If it truly is the Q-function, then that implies that \gamma = 0 and Equation 4 is meaningless -- there is no TD-Error. Can the authors kindly clarify?

It is unclear how this paper is presenting an approach to "shared autonomy." See this work by Reddy et al. (2018).

Reddy, S., Dragan, A.D. and Levine, S., 2018. Shared autonomy via deep reinforcement learning. arXiv preprint arXiv:1802.01744.

Why are none of the baselines able to match the Neural Expert?

Why is the neural expert not at near 100% performance?

**Relation To Broader Scientific Literature:**

Except for the issue noted below regarding Confidence-based Autonomy, the paper generally covers the recent literature in interactive machine learning for an ML audience. However, the awareness of human-centered literature and decades of work on this topic seems to be lacking. I recommend more thoroughly reading the paper's own references, such as

Argall, B. D., Chernova, S., Veloso, M., and Browning, B. A survey of robot learning from demonstration. Robotics and autonomous systems, 57(5):469–483, 2009.

More recent papers/books might also help:

Esmaeil Seraj, Kin Man Lee, Zulfiqar Zaidi, Qingyu Xiao, Zhaoxin Li, Arthur Nascimento, Sanne van Waveren, Pradyumna Tambwekar, Rohan Paleja, Devleena Das and Matthew Gombolay (2024), "Interactive and Explainable Robot Learning: A Comprehensive Review", Foundations and Trends® in Robotics: Vol. 12: No. 2-3, pp 75-349. http://dx.doi.org/10.1561/2300000081

Ravichandar, H., Polydoros, A.S., Chernova, S. and Billard, A., 2020. Recent advances in robot learning from demonstration. Annual review of control, robotics, and autonomous systems, 3(1), pp.297-330.

Zare, M., Kebria, P.M., Khosravi, A. and Nahavandi, S., 2024. A survey of imitation learning: Algorithms, recent developments, and challenges. IEEE Transactions on Cybernetics.

**Theoretical Claims:**

There are no theoretical claims.

---

> ### Author Rebuttal · Authors · 2025-03-31
>
> Thank you for reading our paper in detail and providing valuable suggestions. We summarize and respond to each question as follows:
>
> __Claims And Evidence:__
>
> >However, the term "sufficient" in the third claim (Line 98) should be softened unless providing a proof.
>
> We revise Line 98: “The expert demonstrations requested by AIM contain corrective actions in safety-critical states, so that they can assist a novice agent to imitate the expert’s policy.”
>
> __Relation To Broader Scientific Literature:__
>
> >The awareness of human-centered literature and decades of work on this topic seems to be lacking. The paper does not discuss Confidence-based Autonomy (CBA).
>
> Thanks for sharing relevant works! We will add these references in the revised version.
>
> __Other Strengths And Weaknesses:__
>
> >The limitations section is quite short and does not fully address the many weaknesses of not doing a real user study.
>
> We will add this to the limitation section: "This paper does not include real-human experiments or user studies, and human demonstrations may be imperfect or faulty."
>
> >Figures 4-5 show unconvincing results regarding the superiority of AIM vs. PVP.
>
> Figures 4 and 5 show that AIM and PVP perform similarly in imitating the expert during safety-critical states and collect high-quality expert demonstrations. In short, __AIM’s intervention rule behaves similarly with the human-gated PVP’s rule, even though AIM is a robot-gated interactive IL method.__
>
> In Figure 4 and 5, we do __not aim to show that AIM’s intervention rule is better than PVP__. According to Line 358-362 in Page 7, since PVP is a human-gated IL algorithm, the key advantage of AIM over PVP is that it requires fewer human cognitive efforts and expert-involved steps.
>
> __Questions for Authors:__
>
> >Under what conditions of f in Equation 2 is the agent able to recover the expert policy?
>
> We clarify in Line 164 (Page 3) that the expert’s intervention method follows Eq. 2:
> $I^{exp}(s, a_r, a_h) = f(a_r, a_h) = \mathbb{I}[\|a_r - a_h\|^2 > \epsilon]$ ($a_r$ is the robot action, $a_h$ is the human action). With sufficient expert demonstrations, we can bound the difference of the value functions of the expert policy and the student’s final learned policy by $\epsilon$ and the horizon $H$.
>
> >Must the f in Equation 2 conform to the equation embedded in the text from Menda et al., 2019 and Hoque et al., 2021a;b?
>
> The function f does not need to conform to Menda et al. (2019) and Hoque et al. (2021). The general form of the intervention is based on the probability of the expert taking the robot’s action, which reduces to Eq. 2 if the expert follows a Gaussian policy. The proof is in Theorem 3.3 of “Guarded Policy Optimization with Imperfect Online Demonstrations”  (Z Xue et al., 2023).
>
> >It is confusing to say that $Q_{\theta}^I(s, a_r)$ is equal to -1 or +1 by assignment (Lines 209-218). That implies that \gamma = 0 and Equation 4 is meaningless -- there is no TD-Error.
>
> In Line 209-218, the proxy value assignment is a learning objective but not a hard constraint to the proxy value function. The AIM loss helps select human preferable actions at those states $s$ with human interventions, and TD loss propagates human preference to the states $s$ __without human interventions__.
>
> >Should it not be the reward function definition for -1 and +1 -- not the Q-function?
>
> We include an ablation study in Fig. 3 of https://limewire.com/d/teXVP#Okg7PsYIne . The figure shows that dropping the TD loss or replacing the Q-labeling by reward-labeling will damage the performance of AIM.
>
> Replacing the Q-labeling by reward-labeling fails in our setting because negative rewards cannot be matched with transitions from unsafe agent actions. In human-involved transitions $(s, a_h, s’)$, we can assign +1 since $s’ \sim P(s, a_h)$ is generated by the human action. However, for dangerous agent actions $a_r$, querying the environment to obtain the resulting state $s’’ \sim P(s, a_r)$ is not feasible.
>
> >It is unclear how this paper is presenting an approach to "shared autonomy." See this work by Reddy et al. (2018).
>
> Thanks for pointing out that our definition of "shared autonomy" differs from Reddy et al. (2018), and we revise the terminology to "__interactive imitation learning__." While Reddy et al. apply human-AI shared control during both training and testing, we only use it in the training phase and evaluate the agent's performance without human involvement in the test phase.
>
> >Why are none of the baselines able to match the Neural Expert?
>
> In Table 1, we require all the baselines to use no more than 2K expert-involved steps. These baselines can match the neural expert with 3.5K expert-involved steps. See Fig 2 of https://limewire.com/d/teXVP#Okg7PsYIne .
>
> >Why is the neural expert not at near 100% performance?
>
> The neural expert is trained using Lagrangian PPO with 20M environment steps. MetaDrive safety environments can present challenging scenarios where a well-trained expert may fail.

---

> > ### Comment · Reviewer_xbPG · 2025-04-07
> >
> > The reviewer appreciates the authors’ response. The clarifications regarding the role of the proxy Q-function and how AIM differs from human-gated methods like PVP, as well as the explanation for why the neural expert does not reach near-perfect performance and why the baselines underperform under constrained expert involvement, were all helpful.
> >
> > The authors' plan to soften the original claim about sufficiency and expand the discussion of prior work, especially with respect to Confidence-Based Autonomy and other foundational literature, is important. The acknowledgment of the limitations around the lack of user studies is a welcome addition. It would have been helpful to see this revision in the actual paper, but ICML does not allow for that.
> >
> > The decision to revise the use of “shared autonomy” to more accurately reflect the scope of the work would improve the paper.

---

> > > ### Author Response · Authors · 2025-04-08
> > >
> > > Thanks again for your thoughtful feedback and constructive suggestions. We will follow your feedback to revise the paper accordingly, especially in adjusting the original claim on sufficiency, enhancing the literature review, and addressing the limitations related to user studies.

---

### Official Review · Reviewer_HDAh · 2025-03-17

**Overall Recommendation:** 3

**Summary:**

The authors proposed Adaptive Intervention Mechanism (AIM), a new robot-gated shared autonomy mechanism that better align agent with human expert thorugh a proxy Q-function. This algorithm requires less human monitoring comparing to human-gated interactive imitation learning methods, while more intelligently and efficiently request human expert intervention comparing to other robot-gated imitation learning methods. The authors tested the algorithms on MetaDrive and MiniGrid Four Room Test and achieved SOTA performance.

**Claims And Evidence:**

See bullet points.

**Essential References Not Discussed:**

See bullet points.

**Experimental Designs Or Analyses:**

See bullet points.

**Methods And Evaluation Criteria:**

See bullet points.

**Other Comments Or Suggestions:**

1. What’s I^{exp}? What’s delta? Though some explanation appears in very later part of the paper, the author should explain each term in the formula (1) in the following paragraph, in order to have a clear background knowledge for the readers.
2. What’s the unit of the y axis for the plots in figure 2? Probability of human taking over?  And how is uncertainty estimated? It seems not quite reasonable that many steps come with 0 uncertainty. Would be great to add clear legends to help better explanation.

**Other Strengths And Weaknesses:**

1. Why L2 distance instead of other distance metrics, for example distributional divergence for action-difference function f? It’s not very clear that in human demonstration data, in same state, a_{h} is unique and deterministic? Especially in task like MetaDrive?
2. Could the authors elaborate more on the choice of fixing the threshold epsilon in (9), I would be interested in knowing if an adaptive switch-to-agent threshold would work and perform even better, or if not better, what could be the reason.
3. Like the clear illustration in figure 2, could the authors give several example under safety-critical states, where other baselines robot-gated IL baselines fail to request for human help but AIM did?
4. The experiments for the continuous action space environment and discrete action space environment seems not well-aligned. Why authors did not compare AIM with Ensemble-Dagger and Thrifty-Dagger for mini-grid?
5. In both table 1 and table 2, results show that AIM is not the one with least total data usage, which is fine, since this not the superior aspect of AIM the authors are claiming. However, in page 7 line 374, the authors are claiming AIM’s mechanism saves training time and the total environment data usage according to figure 4, I think it’s not a fair comparison. (AIM with low rate of requesting human intervention could lead to longer training time in some cases.) If comparing vertically in figure 4, using same amount of data, AIM has the least deviation in the critical states, then you should also limit the experiments in table 1 and 2 to same total data then compare the success rate.
6. Since the authors claim the AIM’s great performance origins from it’s adaptive mechanism, I think if would be helpful to add a direct visualization of how the intervention rate changes during the training/testing stage, whether the agent request fewer human intervention as the agent getting more proficient. This should also be compared with other methods.
7. Since AIM has a human-gated warm up stage, would there be a chance that introducing this warm up stage to Ensemble-DAgger and Thrifty-DAgger would also leads to better performance? Then authors argument would be weakened.

**Questions For Authors:**

See bullet points.

**Relation To Broader Scientific Literature:**

Clear literature review.

**Theoretical Claims:**

See bullet points.

---

> ### Author Rebuttal · Authors · 2025-03-31
>
> Thank you for taking the time to carefully read through and understand our paper, and provide constructive feedback. We summarize and respond to each question as follows:
>
> __Other Strengths And Weaknesses:__
>
> >1.Why L2 distance instead of other distance metrics? Is $a_h$ unique and deterministic in task like MetaDrive?
>
> We use L2 distance because it is simple to implement and effectively identifies states where the agent's actions deviate significantly from expert behavior. Fig 3 in https://limewire.com/d/teXVP#Okg7PsYIne shows that using the L1 distance metric does not affect the performance. In MetaDrive, the expert action $a_h$ is stochastic, as the expert policy is a stochastic policy trained with Lagrangian PPO.
>
> >2.Elaborate more on the choice of fixing the threshold epsilon in Eq. 9.
>
> The threshold $\epsilon$ controls __the length of expert demonstrations after the agent requests help__. We choose $\epsilon$ based on observations from the human replay buffer in warm-up steps (Eq. 8, Line 240). A small $\epsilon$ leads to overuse of expert help, while a large $\epsilon$ results in insufficient corrections, slowing down training.
>
> >Does an adaptive switch-to-agent threshold work and perform better?
>
> An adaptive switch-to-agent threshold does __not__ significantly improve the number of expert-involved steps. In our current setup, the agent requires only 5–10 steps of expert help before returning to self-exploration, which is nearly the minimum needed for safety.
>
> >3.Give several examples under safety-critical states, where other baselines robot-gated IL baselines fail to request for human help but AIM did.
>
> In Fig. 1 of https://limewire.com/d/teXVP#Okg7PsYIne , when the car approaches the road boundaries, AIM successfully requests human intervention, while other uncertainty-based methods output a low uncertainty estimate and fail to signal the need for help.
>
> >4.Why did authors not compare AIM with Ensemble-Dagger and Thrifty-Dagger for mini-grid?
>
> In Page 6 Line 310-314, we mentioned that the two methods rely on the action variance for uncertainty estimation, which doesn’t work well with __discrete action spaces__ like MiniGrid. 'Action variance' refers to the variance of the output actions across the ensemble of policy networks.
>
> To apply the two baselines to MiniGrid, we need to replace their variance-based uncertainty estimations by the __entropy-based__ estimation, which is the entropy of the action distributions derived from the soft Q-value. Table 1 of https://limewire.com/d/teXVP#Okg7PsYIne shows that AIM reduces the expert-involved steps needed compared with the two baselines.
>
> >5.In page 7 line 374, the authors are claiming AIM’s mechanism saves training time and the total environment data usage according to figure 4, but results show that AIM is not the one with least total data usage.
>
> Thanks for pointing out the misleading claim. In Figure 4, we show that AIM requires fewer environment samples than other robot-gated baselines (Thrifty-DAgger and Ensemble-DAgger) to approach expert actions in safety-critical states. In Table 1 and Table 2, we highlight that AIM reduces expert-involved steps and cognitive effort, though it may require more training time. Thus, we revise line 374 to:
>
> "Compared with __other robot-gated IIL baselines__, AIM requires fewer environment data usage to __imitate expert actions in safety-critical states__."
>
> >6.How does the intervention rate change during the training/testing stage?
>
> We visualize the overall intervention rate in the training stage in Fig. 4 of https://limewire.com/d/teXVP#Okg7PsYIne . Our method AIM’s intervention rate matches that of human-gated PVP and is lower than other robot-gated baselines.
> According to Line 281-284, in the test stage, we evaluate the agent’s performance without expert involvement, so there’s no intervention rate during testing.
>
> >7.Does introduce warm up stage to Ensemble/Thrifty-DAgger lead to better performance?
>
> In our experiment (Table 1 and Table 2), we already introduced a warm up stage to all the baselines including Ensemble-DAgger, Thrifty-DAgger, and PVP for fairness. (i.e., demonstrating initial two trajectories as we do for AIM).
>
> __Other Comments Or Suggestions:__
>
> >1.What’s $I^{exp}$, $\delta$ in the formula (1)?
>
> $I^{exp}$ is the human-gated intervention criterion, where the expert decides whether to take over control at state s using human control signal $a_h$ when the agent outputs action $a_r$. The $\delta$ function in Eq. 1 is the __Dirac delta distribution__.
>
> >2.What’s the unit of the y axis for the plots in figure 2? How is uncertainty estimated?
>
> In Figure 2, the y-axis represents the uncertainty estimation: $Var(a_n) - \varepsilon$, where $Var(a_n)$ is the variance of agent actions and $\varepsilon$ is the switch-to-human threshold in Ensemble-DAgger. Human help is requested when $Var(a_n) > \varepsilon$. We plot $\max(0, Var(a_n) - \varepsilon)$ to visualize the timesteps when human help is requested.

---

> > ### Comment · Reviewer_HDAh · 2025-04-03
> >
> > Thank the authors for the detailed response. I have raised my score to 3.

---

### Decision · Program_Chairs · 2025-05-01

**Decision:**

Accept (poster)

**Comment:**

This paper introduces Adaptive Intervention Mechanism (AIM), an imitation learning framework designed to improve the efficiency of human-robot collaboration by adaptively determining when to request expert input. Unlike previous robot-gated shared autonomy methods that rely on high-entropy heuristics, AIM leverages a learned Q-function to predict the necessity of expert guidance based on disagreement between the agent and the expert. This approach enables more selective and efficient intervention, reducing the need for constant human supervision while maintaining high task performance. The method was validated on benchmark environments such as MetaDrive and MiniGrid Four Room, where it achieved state-of-the-art results and demonstrated improved alignment with expert behavior compared to existing human-gated and robot-gated imitation learning strategies.